# Epithelioid Hemangioma of the Spine: A Case Series and Treatment Flow Chart—Experience from a Single Centre

**DOI:** 10.3390/cancers16142557

**Published:** 2024-07-16

**Authors:** Emanuela Asunis, Chiara Cini, Mario De Robertis, Cristiana Griffoni, Stefano Bandiera, Alberto Righi, Riccardo Ghermandi, Valerio Pipola, Marco Girolami, Giuseppe Tedesco, Marco Gambarotti, Alessandro Gasbarrini

**Affiliations:** 1Department of Spine Surgery, IRCCS Istituto Ortopedico Rizzoli, 40136 Bologna, Italy; emanuela.asunis@ior.it (E.A.); chiara.cini@ior.it (C.C.); stefano.bandiera@ior.it (S.B.); riccardo.ghermandi@ior.it (R.G.); valerio.pipola@ior.it (V.P.); marco.girolami@ior.it (M.G.); giuseppe.tedesco@ior.it (G.T.); alessandro.gasbarrini@ior.it (A.G.); 2Department of Biomedical Sciences, Humanitas University, 20072 Pieve Emanuele, Italy; mario.derobertis@humanitas.it; 3Department of Neurosurgery, IRCCS Humanitas Research Hospital, 20089 Rozzano, Italy; 4Department of Pathology, IRCCS Istituto Ortopedico Rizzoli, 40136 Bologna, Italy; alberto.righi@ior.it (A.R.); marco.gambarotti@ior.it (M.G.); 5Department of Biomedical and Neuromotor Sciences, Alma Mater Studiorum University of Bologna, 40126 Bologna, Italy

**Keywords:** epithelioid hemangioma, intralesional surgery, en bloc resection, pre-operative embolization, treatment flowchart

## Abstract

**Simple Summary:**

Epithelioid vascular tumors are rare bone vascular lesions with a spectrum of overlapping cancer entities with varying degrees of malignant potential that remain controversial because of their rarity, unusual morphologic features, and unpredictable biologic behavior. The classification of vascular tumors of bone proposed by the World Health Organization in 2020 incorporated morphologic findings with available genetic evidence, including mainly angiosarcoma, epithelioid hemangioma, pseudomyogenic hemangioendothelioma, and epithelioid hemangioendothelioma. Here, we report our experience with eleven patients suffering from epithelioid hemangioma (EH) of the spine. As very few cases of EH of the spine have been reported in the literature, the evidence-based decision-making process for these patients can be very difficult. Thus, we propose a treatment flowchart based on our single center’s experience.

**Abstract:**

Epithelioid hemangioma is recognized by the World Health Organization as a distinct benign neoplasm; however, it is characterized by locally aggressive and rarely metastasizing behavior. Epithelioid vascular tumors are rare bony vascular lesions with varying degrees of malignant potential that remain controversial because of their rarity, unusual morphological features, and unpredictable biological behavior. The application of new molecular tools, such as massive parallel sequencing technologies, have provided new diagnostic markers and an opportunity to further refine the classification of bone vascular neoplasms. Very few cases of EH of the spine have been reported in the literature; therefore, it is difficult to make evidence-based therapeutic decisions for these patients. We report herein our experience with eleven patients suffering from EH of the spine. The study population included three males and eight females treated in our center from 2016 to the present; the average age was 44.8 years (range 14–75 years). The surgical, clinical, and radiographic data were retrospectively analyzed. The mean follow-up was 34.8 months. All patients presented lytic vertebral body lesions, six of them with pathological fracture. The majority of patients (80%) presented myelo-radicular compression. All patients were surgically treated, and preoperative embolization was performed in all cases. In light of the literature review and the clinical experience of our center, we can consider EH a locally aggressive tumor that requires surgical treatment in case of symptoms. Here, we propose a treatment algorithm that could be useful in the management of patients with this rare disease.

## 1. Introduction

Epithelioid vascular tumors are rare bone vascular lesions with a spectrum of overlapping cancer entities with varying degrees of malignant potential that remain controversial because of their rarity, unusual morphologic features, and unpredictable biologic behavior. The pathologic classification of the epithelioid profile of vascular tumors has been a source of controversy [1]. The application of new molecular tools, such as massive parallel sequence technologies, has provided new diagnostic markers, including immunohistochemical markers (FOSB, CAMTA1, YAP1, and MYC), recurrent gene fusions (e.g., those involving FOS, FOSB, YAP1, and WWTR1), and the opportunity for further refinements of the classification of bone vascular neoplasms [1,2,3,4,5]. Thus, it is currently proposed that osseous vascular tumors have to be classified as their soft tissue counterparts, therefore avoiding organ-based diagnostic discrepancies [1,6]. This classification of vascular tumors of bone proposed by the World Health Organization’s (WHO) classification of soft tissue and bone tumors in 2020 incorporated morphologic findings with available genetic evidence, including mainly angiosarcoma, epithelioid hemangioma, pseudomyogenic hemangioendothelioma, and epithelioid hemangioendothelioma. Epithelioid hemangioma (EH) is classified as a benign tumor according to the 2018 ISSVA classification, and the causal gene for vascular abnormalities is recognized to be FOS [7]. The use of the ISSVA classification implies the standardization of diagnosis and treatment because it is a fundamental and internationally accepted classification system for vascular abnormalities. Therefore, the molecular pathology of vascular anomalies and the evaluation of the most recent results on genetic abnormalities are described in reference to the 2018 ISSVA classification. Indeed, EH is recognized as a distinct neoplasm featuring endothelial differentiation, most frequently encountered in soft tissues. The second-most typical site for this tumor is bone [8]. Immunohistochemically, epithelioid hemangiomas are characterized by the strong expression of vascular markers (CD31 and ERG); the variable expression of FOS and/or FOSB; and negativity for pan-keratin AE1/AE3 (typical of pseudomyogenic hemangioendothelioma), CAMTA1 and TFE3 (peculiar of epithelioid hemangioendothelioma), and MYC (found in some cases of epithelioid angiosarcoma) [9,10]. In the setting of FOS fusions in EH, rearrangement results in protein truncation with the loss of its transactivation domain. This leads to decreased susceptibility to proteosomal degradation of FOS protein and the subsequent deregulated growth of endothelial cells. In contrast, the downstream events of FOSB rearrangements in EH are less clearly elucidated, but the fusion oncoprotein retains the critical DNA binding motif and the transactivating domain, resulting in FOSB mRNA transcriptional upregulation [5,7]. The diagnosis of epithelioid hemangioma was supported by molecular analyses (FISH or RT-PCR) whenever the quality and quantity of biological material were adequate [1,5,11,12].

Despite a precise morphological, immunohistochemical, and molecular definition of the tumor entity, epithelioid hemangioma remains a highly controversial entity in terms of clinical behavior. Indeed, the WHO classification of soft tissue and bone tumors classifies epithelioid hemangiomas of soft tissues as benign tumors, whereas bone epithelioid hemangiomas are considered intermediate behavior/locally aggressive tumors due to their common multifocal presentation and potential local destructive growth [9].

The long tubular bones of the extremities are the most common sites for EH in osseous tissue, followed by the flat bones and the short tubular bones of the distal lower extremities [13]. EH of the spine is reported to represent 6–16% of osseous EH [8]. Very few cases of EH of the spine have been reported in the literature; therefore, the evidence-based decision-making process for these patients can be very difficult. We report herein our experience with eleven patients suffering from EH of the spine.

## 2. Materials and Methods

The data for this study were collected retrospectively from the electronic health records of patients surgically treated from 2016 to 2023 at the Istituto Ortopedico Rizzoli. Patients provided informed consent for an observational study (registry) approved by the local Ethics Committee on December 14th, 2016 (protocol number 0022814).

All patients who had a diagnosis of EH with spinal localization and a minimum follow-up period of 6 months with complete clinical and radiological data were included in the study.

Nine of our cases were examined by CT-guided needle biopsy, and the diagnosis was confirmed by senior musculoskeletal pathologists. Only one case was treated in an emergency setting. All clinical, radiographic, and histological data available were collected. All imaging was examined. The vertebral lesions were characterized in terms of topographic distribution, number of levels involved, and elements involvement in the vertebral compartment. Type and duration of symptoms at admission (or baseline) were recorded.

The individual patients’ performance statuses were assessed through ECOG score at hospital admission. The diagnosis of all patients was made by morphological, immunohistochemical, and molecular analysis when feasible due to specimen quality (i.e., sample decalcification) (see Figure 1 for an example of histological examination). The treatment strategy was assessed globally, considering all procedures complementary to the primary surgery (selected arterial embolization (SAE), radiotherapy (RT)). Post-operative pathological margin assessment was matched with surgeon’s pre- and intraoperative impressions. All the adverse events were systematically reported and graded according to SAVES v-2 [14].

Moreover, we conducted a narrative review of all cases of EH of the spine. We searched on Pubmed with the keywords ‘‘epithelioid hemangioma” and “vertebral” or “spine’’ to include only papers concerning EH of the spine.

## 3. Results

The study population included eleven patients: three males and eight females. The average age of patients was 44.8 years (range 14–75 years). All patients presented lytic vertebral body lesion, six of them with pathological fracture. The majority of patients (80%) presented myelo-radicular compression. All patients were surgically treated, and preoperative embolization was performed in all cases. The demographic and clinical data are reported in Table 1, while data concerning treatments and follow-up are reported in Table 2. WBB staging [15] and post-operative pathological margins were evaluated (Table 1 and Table 2).

### 3.1. Case 1

A 29-year-old woman was referred to our department in EM suffering one-week acute back pain. A computed tomography scan demonstrated a T5 pathological fracture (vertebra plana), two lytic lesions in the vertebral body and posterior arch of T4, and one lesion in the T6 body.

The tumor extended from the left neuroforamen at T4-T5 to the costotransverse joint. A CT-guided biopsy confirmed the diagnosis as EH.

Considering the intensity of pain and the vertebral instability, the patient underwent en bloc resection surgery through a posterior midline approach after pre-operative embolization. Anterior column reconstruction of the thoracic spine was performed with a 3D-printed titanium cage. Six months after the primary surgery, the patient underwent segmental resection of the left fifth rib (local recurrence). At 44 months after the thoracic surgery, the patient presented local costal recurrence and underwent a wide resection. No evidence of disease (NED) was detected at the last follow-up (85 months) in the absence of symptoms. Radiographic images concerning the pre-operative and post-operative examinations are reported in Figure 2, Figure 3, Figure 4 and Figure 5.

### 3.2. Case 2

A 70-year-old female reported having occasional low back pain without trauma for about 4 years. A CT scan of the lumbar spine showed a lytic expansive lesion of T11 with right extension and involvement of the omolateral 11° rib. After 2 years from the onset of pain, the patient went to another hospital where an incisional biopsy was performed with a histological diagnosis of epithelioid hemangioma. She was subsequently admitted to our institution. SAE and CT-guided trocar biopsy were performed to confirm the previous diagnosis. Then the patient underwent wide en bloc resection by single-stage double approach with a segmental resection of the affected right rib. An osteolytic lesion involving the left 10th rib at the level of the costovertebral junction was also observed, associated with a soft tissue component. The anterior column was reconstructed with a bone allograft.

A deep wound infection was observed in the early postoperative period and treated by a surgical debridement. After 17 months, a mechanical complication occurred with loss of correction, requiring revision and the extension of instrumentation to the pelvis. After 24 months, a vertebroplasty at the level above the upper instrumented level (UIV) was necessary. At 32 months of follow-up, no evidence of disease (NED) was detected, even though the patient complained of low back pain and left radiculopathy.

### 3.3. Case 3

A 16-year-old woman was admitted to our department suffering from a 2-month history of low back pain due to a cord compression caused by a T10 osteolytic lesion, confirmed by a CT scan and MRI assessment, involving the vertebral body. The spinal cord was compressed by the intracanalar component. She underwent an en bloc resection by posterior approach following a pre-operative SAE; the anterior column was reconstructed with a carbon fiber prosthesis. In the post-operative course, the patient had a pneumothorax treated without sequelae. At her last follow-up (72 months), no evidence of local recurrence (NED) or symptoms was reported.

### 3.4. Case 4

A 14-year-old male was referred to our department due to acute back pain. He had no sphincteric symptoms, but neurologic signs included the presence of mild hyposthenia of the left psoas 3/5 according to Medical Research Council strength scale, paresthesia at the left L2 nerve root, left positive Babinski, and hyperactive deep reflexes. He had no difficulty in standing or walking without support and could climb the stairs without the assistance of a handrail. He accessed the emergency unit for acute severe back pain onset. A CT scan of the lumbar spine showed a multilocular lytic lesion at L2 mainly in the posterior elements, pedicles, and transverse processes. MRI scans of the lumbar spine demonstrated a diffusely homogeneous enhancing spinal tumor with low intensity on the T1-weighted MRI scan and hyperintensity on the T2-weighted MRI scan. The tumor extended into the spinal canal and compressed the spinal cord at L2 level. A CT-guided trocar biopsy and SAE were performed. After the pathologist’s results, because of the local aggressiveness of the EH, another SAE procedure was performed. The patient was submitted to an intralesional excision combined with a posterior L1-L3 stabilization with CRF-PEEK instrumentation and an anterior column reconstruction by PMMA. One month after the surgery, the patient was submitted to 40 Gy radiotherapy. At the last follow-up, 6 months after surgery, the patient was without pain or evidence of local recurrence (NED).

### 3.5. Case 5

A 50-year-old woman presented with a 6-month history of neck pain and left arm radicular pain without focal deficits or myelopathy. Imaging (CT scan and MRI) demonstrated a lytic lesion in the C6 vertebra. The tumor extended into the left neuroforaminen at C6. A CT-guided trocar biopsy confirmed the diagnosis of EH. The tumor was embolized preoperatively, and the patient underwent a C6 corporectomy. After intraoperative diagnostic evaluation, a curettage at the level of the lower portion of the body of C5 was also performed. Arthrodesis was performed with non-autologous fibular bone graft and C5-C7 carbon fiber plate. At the last follow-up (52 months), the patient was in good general condition and without symptoms or local recurrence (NED).

### 3.6. Case 6

A 48-year-old woman presented to the emergency department with acute thoracic back pain. The CT scan of the thoracic spine revealed a pathological fracture of T6 and an osteolytic lesion with a partially sclerotic border involving the T6 vertebral body, the right T6-T7 neuroforamen, and the omolateral posterior arch. The MRI scan showed a pathologic fracture with spinal cord compression. The patient underwent an intralesional excision and posterior stabilization. The pathologist’s diagnosis of EH was confirmed, and the patient underwent adjuvant conventional RT. After 52 months, the patient showed no signs of recurrence. The last imaging assessment showed no recurrence, adequate spinal alignment, and no hardware failure. The patient was in a good clinical condition and presented only occasional episodes of cervicalgia, which resolved spontaneously without the need for medication (NED).

### 3.7. Case 7

A 39-year-old woman presented with a one-year history of low back pain without neurologic signs or symptoms. The patient underwent right proximal humerus resection and reconstruction with an osteoarticular autograft for EH in 2001. In 2014, she underwent a curettage of the right proximal radius with a diagnosis of EH. In 2015, a craniotomy was performed for the frontal bone localization of EH. In 2018, she underwent a curettage for the recurrence of EH of the right humerus. In 2019, a curettage was performed for the recurrence of EH at the right proximal radius. In 2020, the patient presented with low back pain. CT and MRI scans demonstrated a lytic lesion in the sacrum. The tumor had low-to-moderate signal intensity on the T1-weighted images and high signal intensity on the T2-weighted images. A CT-guided biopsy confirmed the diagnosis as EH. The patient underwent an intralesional excision of the sacral lesion. In 2021, the patient was diagnosed with EH of the distal right humerus. The patient was surgically treated, and the loss of substance was replaced with synthetic resin and a homoplastic bone splint. The construction was then held in place with special instrumentation and opposing screws. In 2022, a new curettage was performed for right humerus recurrence. In July 2023, a new localization in the proximal right radius was detected. The patient underwent a new curettage of the lesion. At the last follow-up, at 41 months after surgery, the patient was without pain or evidence of sacral recurrence.

### 3.8. Case 8

A 53-year-old man presented with a 5-month history of back pain. A lesion was suspected on the plain radiographs. An MRI scan revealed increased signal intensity on the T2-weighted images focally on the anterior vertebral body of T12 and both the pedicles without soft-tissue extension and minimal impingement of the dural sac, without spinal cord displacement. A CT scan demonstrated a lytic lesion within the vertebral body of T12 and the involvement of both pedicles. A CT-guided biopsy was performed. The patient underwent preoperative embolization followed by a double surgical approach en bloc resection with planned focal intralesional margin transgression of T12. Reconstruction by a carbon fiber expandible cage and T9-L2 posterior stabilization were performed. The postoperative course was complicated by a left pneumothorax that required a chest drain procedure. At the last follow-up, 24 months after surgery, he had no local recurrence (NED), but he continues to complain of pain in his lower thoracic spine.

### 3.9. Case 9

A 61-year-old woman presented with a 2-month history of low back pain and the new onset of right sciatica. An MRI scan revealed a lesion in L2 (pathological tissue with involvement of the isthmus bilaterally, paravertebral muscles, posterior arch, with spinous process of L2). In 2023, after biopsy diagnosis and pre-operative SAE, the patient underwent an intralesional excision and posterior stabilization (L1 and L3). To give more stability to the construction, a vertebroplasty was performed at T12 and L4. At 6 months of follow-up, the patient was without recurrence of L2 lesion and was pain free (NED).

### 3.10. Case 10

A 76-year-old man presented with a 2-year history of low back pain and left sciatica. A CT-guided biopsy was performed at another institution, and a new revision of the histopathological samples was carried out by our Pathological Anatomy Department, which did not allow a diagnosis to be made with certainty. Thus, a new CT-guided biopsy of L2 was performed, and an EH diagnosis was confirmed. The patient underwent SAE and then an intralesional excision of the lesion. The structure was then stabilized with augmented screws and vertebroplasty of the vertebrae adjacent to the instrumentation. At the last follow-up (7 months), the patient had no evidence of local recurrence or new localization (NED).

### 3.11. Case 11

A 30-year-old man presented with dorsal pain. The MRI demonstrated a lytic lesion involving T7 left pedicle and posterior arch with ESCC. A CT-guided biopsy was performed in another center, and our histological revision confirmed an EH diagnosis. The patient underwent intralesional excision and posterior stabilization. The postoperative process was complicated by pleural empyema, and a thoracoscopy for pleural cavity drainage was required. At the sixth month of follow-up, the control MRI revealed a new lesion in the superior left body of T1, and a new CT-guided biopsy was performed with a non-diagnostic outcome. It was decided to continue with a longitudinal follow-up, performing an MRI scan every 4 months.

## 4. Discussion

EHs are rare tumors. The main involvement is at soft tissue level (skin and subcutaneous tissue of the head and neck region). Bone localizations are rare: only 19 cases of EH localized in the spine are reported in a systematic review by Okada et al. [16]. Although the WHO describes them as benign tumors, their potential for local recurrence and aggressiveness are not deeply known. Moreover, vertebral lesions are frequently associated with instability, pathological fractures, and the compression of neural structures [8]. Epithelioid hemangioma in soft tissue is considered clinically benign, whereas in bone, it behaves as a locally aggressive, rarely metastasizing lesion and is, therefore, considered as an intermediate entity. However, they are effectively the same tumoral entity [1,5,11,12].

In the literature, some case reports are available, describing en bloc and intralesional excision treatments of individual patients affected by EH. Ling et al. [17] reported on a lesion that was resected en bloc from the underlying dense bone, and anterior reconstruction was performed using an iliac bone graft [17].

Boyaci et al. [18] reported a case series of six patients affected by EH of the spine and underlined how these tumors can cause instability and neurological symptoms due to neural-element compression. In particular, in their study, five out of the six patients needed decompressive surgery. Of these, two underwent en bloc resection, requiring no further treatment, while three patients were candidates for radiation therapy following intralesional excision.

In our series, one patient presented multiple bone lesions. All eleven patients underwent surgery limited to the vertebral level affected by pathological fracture or spinal cord compression. Before surgery, all patients underwent selective arterial embolization (SAE) of the lesion. We believe that it is useful to perform a preoperative embolization as it can reduce intraoperative bleeding, in particular when an intralesional excision is planned.

Four patients underwent an en bloc resection, and seven patients underwent an intralesional excision. In our population, one case of local recurrence at costal level was detected after intralesional excision and surgically treated.

Nielsen et al. [8] reported local recurrence in 4 out of 36 patients (11%) with bone EH during follow-up. One of these patients also presented the involvement of a regional lymph node. Only one patient was treated in an emergency setting with decompression and fixation for a pathological fracture of unknown nature. After the surgery, radiation therapy was performed in order to improve the local control of the disease. No local recurrence or disease progression were observed in these cases [8].

Other adjuvant treatments (in particular, pazopanib and radiofrequency) have been proposed in the literature for rare cases of bone EH showing multiple localization or metastatic behavior [19,20].

The correct treatment of EH is not yet clear, principally because the small size of the population affected limits high-quality evidence-based strategies. However, analyzing the literature and the data from this case series, which is the largest in the literature, we can assert that the treatment of choice for this tumor is en bloc resection when anatomically and clinically possible. In some cases, the combination of intralesional resection and radiotherapy can be performed if en bloc resection with wide/marginal margins is associated with unacceptable morbidity and loss of function because of tumor extension or anatomical constraints. This choice of treatment can be justified by the unknown and unclarified behavior of EH. Radiation treatment alone should be limited to patients who are not eligible for surgery in the absence of vertebral mechanical instability or signs of neural compression. A single case of vertebral localization treated with conventional RT is cited in the literature. In this clinical case, described by Siltumens et al. [21], it is highlighted that radiotherapy treatment can have good efficacy in pain control. However, the study also demonstrates that radiotherapy treatment alone is not curative, but it can be effective in local control, especially in patients who cannot undergo surgery [21]. This further validates our line of treatment, even if studies with a high level of evidence comparing different types of RT protocol as a primary treatment and different surgical strategies or the extent of resection are not available yet.

We are aware of the limited number of patients in our study population; however, few data are reported in the literature concerning epithelioid hemangioma of the spine, and our case series is currently the largest worldwide. Moreover, a comparison with the literature allowed us to draw initial conclusions that we hope could be a starting point for further investigations of this still poorly understood condition.

Therefore, we decided to propose a flowchart for the treatment of EH based on the literature evidence and on the case series of our institution (Figure 6).

The proposed treatment algorithm starts from the correct histo-molecular characterization of EH. The most frequent clinical onset of a patient affected by vertebral EH is axial pain that is mainly mechanical. Other symptoms are neurological impairments deriving from direct compression of the spinal cord or nerve roots. Otherwise, compression can be caused by a pathological fracture.

The first step of the algorithm is the anesthesiologic evaluation of the patient’s eligibility for surgery: if the patient is inoperable for comorbidities, non-surgical options will be considered (RT or SAE). These treatments do not have curative intent but represent palliative therapies (aimed to control pain first) in case the patient is not considered operable.

If the patient is considered operable, the extent of epidural spinal cord compression (ESCC) and the severity of the neurological damage are assessed. If an acute neurological deficit is present with the possibility of neurological recovery, emergency surgery by intralesional excision and stabilization is advisable. If there is no possibility of neurological recovery, RT or SAE should be considered to reduce pain, in case of undetectable signs of instability at pre-operative imaging assessment. When there is no neurological deficit or pain, the impending pathological fracture and risk of instability are thus evaluated by SINS score [22].

If careful preoperative planning demonstrates its feasibility, in particular in those cases where pain or a recoverable neurological deficit are present and the risk of instability or an impending pathological fracture are defined, wide or marginal en bloc tumor resection should be performed, preceded by SAE. When en bloc resection is not feasible, an intralesional excision and stabilization should be considered in case of instability or curable neurological impairment. After intralesional excision, adjuvant therapy such as radiotherapy should be contemplated (among other options). If no risk of instability or impending pathological fracture is defined and the patient has no pain or neurological deficit, a clinical and radiological outpatient observation should be considered. If the lesion increases and causes neurological deficit or instability, surgery should be considered as the first option.

## 5. Conclusions

This study reports the radiological and clinical outcomes of a series of eleven patients surgically treated for epithelioid hemangioma of the spine in a single center and proposes a treatment flowchart that could be helpful for decision-making and the management of patients affected by this rare pathology.

## Figures and Tables

**Figure 1 cancers-16-02557-f001:**
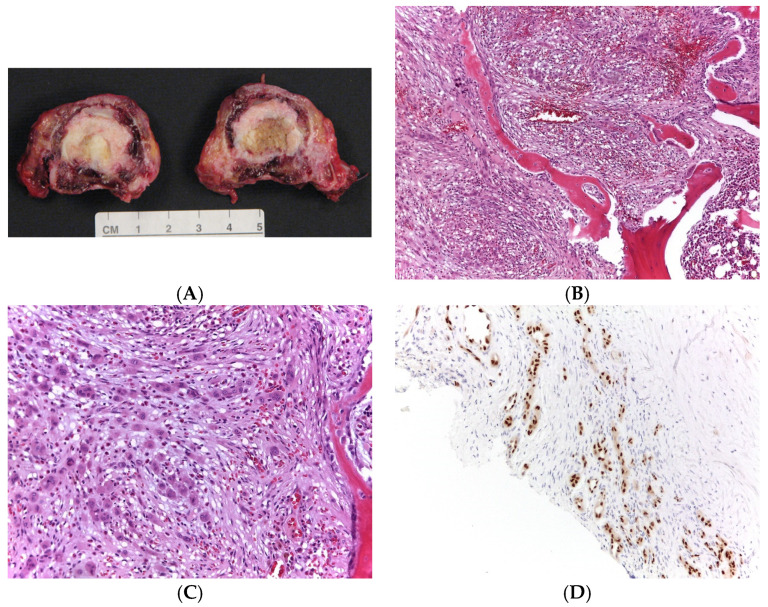
(**A**): Gross specimen relative to en bloc resection; (**B**,**C**): Morphologically, a prominent proliferation of small, capillary-sized vessels, sometimes lacking a well-defined lumen lined by epithelioid endothelial cells with an enlarged nucleus. Occasional eosinophils and lymphocytes are evident (hematoxylin and eosin staining, (**B**): 100× of magnification, (**C**): 400× of magnification). (**D**): These neoplastic endothelial cells show a strong nuclear expression for FOSB antibody (200× of magnification).

**Figure 2 cancers-16-02557-f002:**
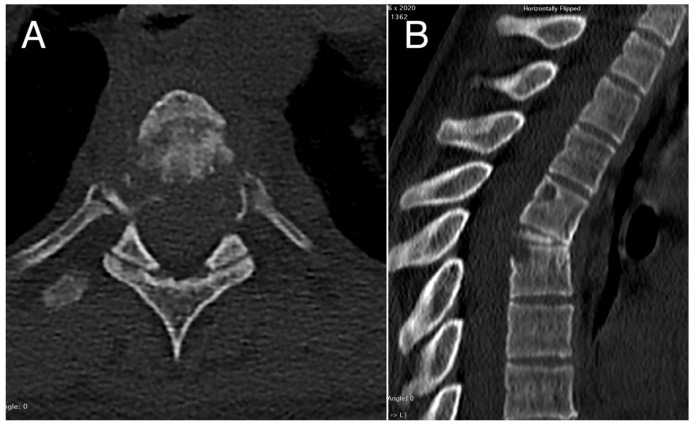
Pre-operative axial (**A**) and sagittal (**B**) CT scans showing the pathological collapse of T5 vertebral body with osteolysis affecting also the corresponding pedicle regions, the adjacent upper portion of T6 vertebral body, and the posterior-upper angle of the T4 vertebral body.

**Figure 3 cancers-16-02557-f003:**
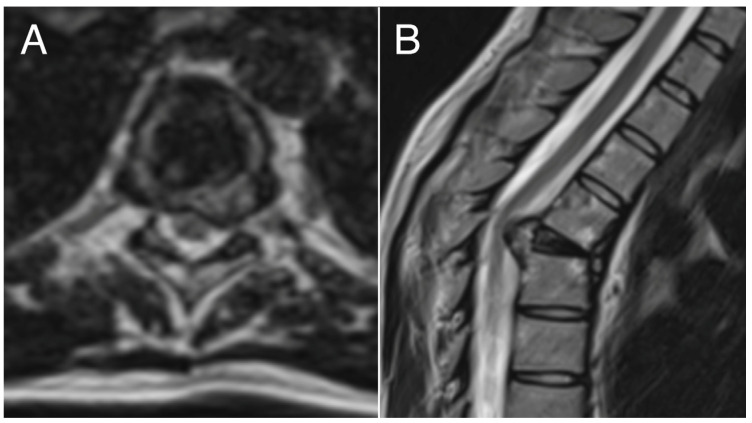
Pre-operative axial (**A**) and sagittal (**B**) MRI examination showing pathologic vertebra plana (T5) and spinal cord compression.

**Figure 4 cancers-16-02557-f004:**
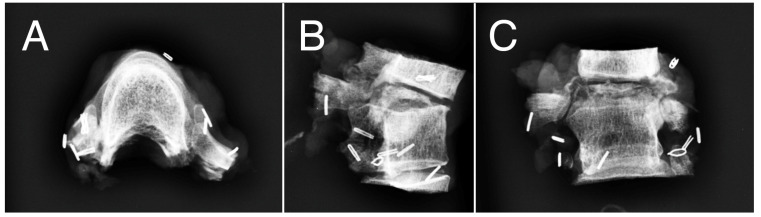
Intraoperative anatomical piece showing resection of vertebra plana T5, adjacent vertebra T6, and partial resection of upper and lower vertebrae with wide resection margins. (**A**) axial view; (**B**) sagittal view; (**C**) coronal view.

**Figure 5 cancers-16-02557-f005:**
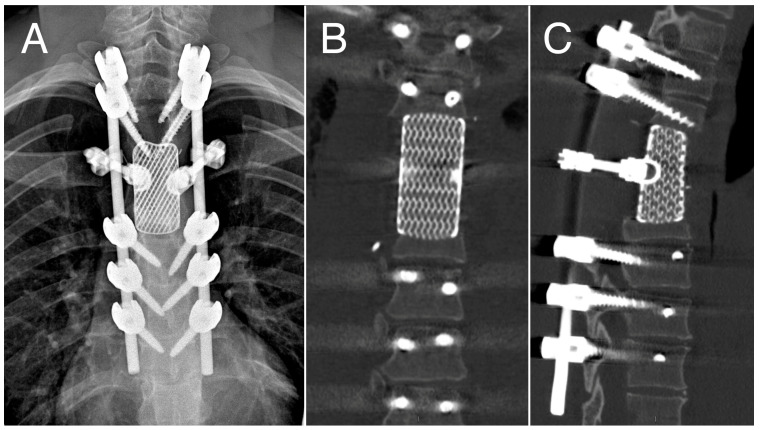
12-month follow-up: Antero-posterior X-ray (**A**) and CT scans (**B**,**C**) showing outcomes of vertebrectomy and reconstruction with custom-made prosthesis.

**Figure 6 cancers-16-02557-f006:**
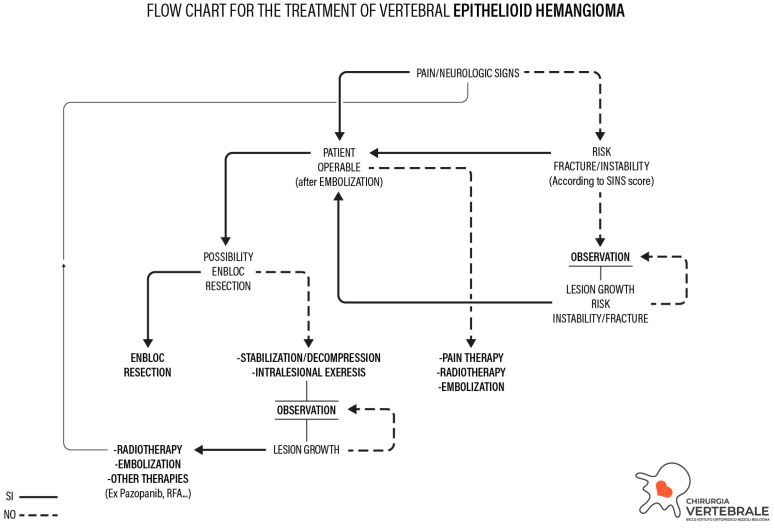
Flowchart for the treatment of epithelioid hemangioma.

**Table 1 cancers-16-02557-t001:** Demographic and preoperative data.

Patient	Sex	Age	Level	Vertebral Compartment Location	Other Localizations	SYMPTOMS (1 Myelopathy, 2 Radiculopathy, 3 Conus/Cauda, 4 None)	WBB Staging
1	F	29	T5	Vertebral body, left costo-trasversary	5th left rib (LR at 6 months FU)	4	Layer C, Sector 4, 5, 8
2	F	70	T11	Vertebral body, right costo-trasversary	11th right rib	4	Layer A, Sector 8, 9, 10
3	F	16	T10	Vertebral body	None	4	Layer C, Sector 6, 7
4	M	14	L2	Posterior elements and pedicles	None	2	Layer D, Sector 4, 3, 2, 1, 12, 11
5	F	50	C6	Vertebral body	None	2	Layer C, Sector 6, 7
6	F	48	T6	Vertebral body and right posterior arch	None	2	Layer C, Sector 6, 7, 8, 9, 10
7	F	39	Sacrum	Sacral promontory	Proximal and distal right humerus, proximal right radius, cranial theca	4	Sacral lesion
8	M	53	T12	Vertebral body and both pedicles with extention to soft tissue	None	2	Layer C, Sector 4, 5, 6, 7, 8, 9
9	F	61	L2	Posterior arch, spinous process	None	4	Layer C, Sector 11, 10, 9
10	F	76	L2	Left vertebral body, left pedicle	None	2	Layer C, Sector 3, 4
11	M	30	T7	Left pedicle and posterior arch	None	1	Layer C, Sector 2, 3, 4

**Table 2 cancers-16-02557-t002:** Treatments and follow-up.

SAE	RT	Surgery	Post-Operative Pathological Margins	Status at Last FU	FU (months)	Adverse Events
Yes	No	En bloc resection	Wide margins	NED	85	None
Yes	No	En bloc resection and segmental resection of affected right rib	Wide margins	NED	32	Wound infection and mechanical failure with loss of correction
Yes	No	En bloc resection	Wide margins	NED	72	Pneumothorax
Yes	Yes	Intralesional excision and posterior stabilization	Planned focal intralesional margins	NED	6	None
Yes	No	Intralesional excision with anterior approach and fixation	Wide margins	NED	52	None
Yes	Yes	Intralesional excision and posterior stabilization	Planned focal intralesional margins	NED	52	None
Yes	No	Intralesional excision	Planned focal intralesional margins	NED	41	None
Yes	No	En bloc resection with planned focal intralesional margins transgression	Planned focal intralesional margins	NED	24	Pneumothorax
Yes	No	Intralesional excision and posterior stabilization	Planned focal intralesional margins	NED	6	None
Yes	No	Intralesional excision and posterior stabilization	Planned focal intralesional margins	NED	7	None
Yes	No	Debulking and stabilization	Planned focal intralesional margins	NED	6	None

## Data Availability

Data supporting reported results are stored in the electronic medical records of the hospital and may be requested from the corresponding author.

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
