# Peer review of "Epithelioid Hemangioma of the Spine: A Case Series and Treatment Flow Chart—Experience from a Single Centre"

_cancers, 2024, doi:10.3390/cancers16142557_

Round 1

Reviewer 1 Report

Comments and Suggestions for Authors

Dear authors,

I read with great interest your manuscript and I think it is valuable and worth to be published. Malignant, borderline vascular or locally aggressive vascular tumors of the bone are very rare, require multidisciplinary diagnostic and treatment teams, and usually lack guidelines and management protocols. That is why special cases must be reported and I suggest a broader approach in the future, for example a systematic review or a meta-analysis.

Regarding this manuscript, I have some comments and recommendations, as follows:

-Abstract: I consider it necessary to introduce a sentence in the abstract regarding the tumoral pathology of EH.

-Introduction: Please also refer to the ISSVA classification of vascular anomalies in the Introduction chapter.

-Materials and methods: I think that more data about the inclusion or exclusion criteria of patients are needed.

-Results: Considering that the journal in which you want to publish (Cancers) has a certain specificity, more details of morphology and pathology are necessary, for each individual patient; also the type of tumor, classification, extent, tumor grade, etc.

Discussions: a subchapter with limitation of the study is needed. Please highlight the particularity and importance of your study as well as the opportunities that this study opens up.

References: More new references are needed. 

Author Response

REVIEWER 1

Dear authors,

I read with great interest your manuscript and I think it is valuable and worth to be published. Malignant, borderline vascular or locally aggressive vascular tumors of the bone are very rare, require multidisciplinary diagnostic and treatment teams, and usually lack guidelines and management protocols. That is why special cases must be reported and I suggest a broader approach in the future, for example a systematic review or a meta-analysis.

Regarding this manuscript, I have some comments and recommendations, as follows:

-Abstract: I consider it necessary to introduce a sentence in the abstract regarding the tumoral pathology of EH.

Considerations about the tumoral pathology of EH have been added in the Abstract, as suggested, and marked in red. 

-Introduction: Please also refer to the ISSVA classification of vascular anomalies in the Introduction chapter.

We added the reference to ISSVA classification in the Introduction, as suggested (marked in red).

-Materials and methods: I think that more data about the inclusion or exclusion criteria of patients are needed.

We added a sentence in Materials and Methods section about inclusion and exclusion criteria (marked in red).

 -Results: Considering that the journal in which you want to publish (Cancers) has a certain specificity, more details of morphology and pathology are necessary, for each individual patient; also the type of tumor, classification, extent, tumor grade, etc.

More details about the pathology for each individual patient have been added in the Tables and two tables are present now (Table 1 and Table 2), which are cited in the Results section.

-Discussions: a subchapter with limitation of the study is needed. Please highlight the particularity and importance of your study as well as the opportunities that this study opens up.

A paragraph highlighting the limitations of the study but also its relevance as the largest case series of spine EH has been added in the Discussion section.

-References: More new references are needed.

Five more references have been added (marked in red). 

Reviewer 2 Report

Comments and Suggestions for Authors

Review

Dear Editor,

Thank you for giving me an opportunity to review this article about Epithelioid Hemangioma of the spine.

It may be worth publishing but I think that the authors have some points to discuss.

Major points:

1.      The authors show good clinical outcome that no evidence of local recurrence in all 11 patients. However, as previously reported, local recurrence rate is not so small. What is the difference between in your methods and some previous papers? Because the authors shows in Table 1, intralesional resection was done in 7 of 11. Why local recurrence rate is 0 in your cases? This may be helpful for spine or other orthopedic surgeons. Description of tips and pitfalls is preferred.     

2.      Given the difficulty of surgery because of the location near spinal cord, lung and etc. Adjuvant procedure may be benefit in the treatment for locally aggressive Epithelioid Hemangioma. SAE was conducted in all 11 cases but I think this SAE is done for bleeding control. What do you think of the role of SAE in order to control local recurrence?

3.      Similarly, how about Denosumab as adjuvant or salvage?

4.      Similarly, Pazopanib?

5.      Similarly, RFA?

6.      In the flowchart, what methods should be applied for risk of fracture or instability? SINS score? Description is required.

7.      Taking abovementioned points into consideration, I would like the authors to correct the flow chart as shown in Figure 6.

Minor points:

1.      Page8, case 11 symptom – Yes. What type of symptom? 1-4?

Author Response

REVIEWER 2  

Dear Editor, 

Thank you for giving me an opportunity to review this article about Epithelioid Hemangioma of the spine.

It may be worth publishing but I think that the authors have some points to discuss.

Major points:

1.The authors show good clinical outcome that no evidence of local recurrence in all 11 patients. However, as previously reported, local recurrence rate is not so small. What is the difference between in your methods and some previous papers? Because the authors show in Table 1, intralesional resection was done in 7 of 11. Why local recurrence rate is 0 in your cases? This may be helpful for spine or other orthopedic surgeons. Description of tips and pitfalls is preferred.  

For each individual case we performed pre-operative embolization, surgical interventions and adjuvant treatments in some cases as reported in the Results section. We observed only one case of local recurrence at costal level during the follow up period; similar results have been obtained by Boyaci et al. in their case series of 6 patients affected by EH of the spine. Nielsen et al. reported 4 cases of local recurrence in 36 cases of bone EH (11%), thus we can say that our rate of local recurrence/progression is in line with those reported in the literature.

  1. Given the difficulty of surgery because of the location near spinal cord, lung and etc. Adjuvant procedure may be benefit in the treatment for locally aggressive Epithelioid Hemangioma. SAE was conducted in all 11 cases but I think this SAE is done for bleeding control. What do you think of the role of SAE in order to control local recurrence?

No data are reported in the literature to support the role of SAE in the local control of EH, even if SAE is successfully used to treat Aneurysmal Bone Cyst as benign spinal tumor (Terzi S, et al. Efficacy and Safety of Selective Arterial Embolization in the Treatment of Aneurysmal Bone Cyst of the Mobile Spine: A Retrospective Observational Study. Spine (Phila Pa 1976). 2017 Aug 1;42(15):1130-1138). Preoperative SAE is also recommended for the treatment of aggressive vertebral hemangioma, in addition to decompression and stabilization (when necessary) and followed by radiotherapy in case of recurrence (Kato K, et al. Vertebral hemangiomas: a review on diagnosis and management. J Orthop Surg Res. 2024 May 24;19(1):310). We could hypothesize that blocking the high blood supply can reduce lesion growth in the case of EH, but this is not demonstrated in the literature.

  1. Similarly, how about Denosumab as adjuvant or salvage?

Denosumab is used for the treatment of Giant Cell Tumour (GCT) (Chawla S, Denosumab in patients with giant-cell tumour of bone: a multicentre, open-label, phase 2 study. Lancet Oncol. 2019 Dec;20(12):1719-1729) and also for Aneurysmal Bone Cyst of the spine (Ghermandi R, Denosumab: non-surgical treatment option for selective arterial embolization resistant aneurysmal bone cyst of the spine and sacrum. Case report. Eur Rev Med Pharmacol Sci. 2016 Sep;20(17):3692-5; Evangelisti et al, submitted). However, no data are reported for the treatment of EH with denosumab.

  1. Similarly, Pazopanib?

The use of Pazopanib as salvage therapy has been reported for three cases of Hemangioendothelioma and bone Epitelioid Hemangioma showing multiple localizations and metastatic behavior (Yakobson A, et  al. Epithelioid Hemangioendothelioma and Epithelioid Hemangioma: Pazopanib as a Potential Salvage Therapy. Case Rep Oncol. 2021 Mar 3;14(1):309-317). We haven’t use this kind of treatment, however, we think it can be taken into account as an alternative treatment in cases with multiple localizations and progression of the disease.

The reference has been reported in the Discussion section.

  1. Similarly, RFA?

The use of radiofrequency has been recently reported for the treatment of a case of multifocal EH of the distal tibia /Savvidou O, et al. Epithelioid Hemangioma of Bone: A Rare Vascular Neoplasm. A Case Report and Literature Review. J Long Term Eff Med Implants. 2022;32(4):47-55). However, in vertebral localizations of EH the application of RFA is more complex, especially in  case of large lesions or those that involve structural elements and result in spinal cord compression. Maybe in case of a local recurrence it could be applied.

The reference has been reported in the Discussion section.

  1. In the flowchart, what methods should be applied for risk of fracture or instability? SINS score? Description is required.

SINS score has been used to evaluate the risk of fracture or instability (it has been cited in the Discussion section and reference has been added).

  1. Taking abovementioned points into consideration, I would like the authors to correct the flow chart as shown in Figure 6.

The flow chart in Figure 6 has been modified according to suggestions.

Minor points:

  1. Page8, case 11 symptom – Yes. What type of symptom? 1-4?

Symptom for case 11 has been added in Table 1

Round 2

Reviewer 1 Report

Comments and Suggestions for Authors

Dear authors,

Thank you for your responses. I found your manuscript improved and ready to be published.

Reviewer 2 Report

Comments and Suggestions for Authors

They nicely addressed my concern.